# Performance of Fall Armyworm Preimaginal Development on Cultivars of Tropical Grass Forages

**DOI:** 10.3390/insects13121139

**Published:** 2022-12-10

**Authors:** Marcos V. C. dos Santos, Priscilla T. Nascimento, Maria L. Simeone, Patrick F. Lima, Rosangela M. Simeão, Alexander Auad, Ivênio Oliveira, Simone Mendes

**Affiliations:** 1Campus de Sete Lagoas, Universidade Federal de São João Del Rei, Rodovia MG-424, Km 47, Bairro Indústrias, Caixa Postal 56, Sete Lagoas 35701-970, MG, Brazil; 2Embrapa Milho e Sorgo, Rodovia MG 424, Km 45, Sete Lagoas 35701-970, MG, Brazil; 3Embrapa Gado de Corte, Av. Rádio Maia, 850, Campo Grande 79106-550, MS, Brazil; 4Embrapa Gado de Leite, Av. Eugênio do Nascimento, 610, Juiz de Fora 36038-330, MG, Brazil

**Keywords:** insect–plant interaction, pest management, plant resistance, *Spodoptera frugiperda*

## Abstract

**Simple Summary:**

Integrated crop systems that intercrop grass forages and grain crops are an important strategy used for increasing crop production and improving nutrient cycling and soil health. However, pests such as the larvae of the insect *Spodoptera frugiperda*, which is able to feed on uncountable plant species, may become a huge threat to these systems. There is a great diversity of forage grass species and cultivars available for use in these systems, so we investigated whether the insect performed differently on different forage species. The performance of this pest was good in most plants observed, though the pest performed poorly in *Panicum maximum* ‘Massai’ and ‘Tamani’ cultivars. On other hand, a careful pest management is necessary for *Brachiaria brizantha* ‘Paiaguás’, ‘Marandu’, ‘Xaraés’ cultivars and *Brachiaria ruziziensis*, since these plants are more suitable for the development of the pest.

**Abstract:**

*Spodoptera frugiperda* (J.E. Smith) (Lepidoptera: Noctuidae) is a polyphagous pest species capable of feeding on almost all forage and grain crops, although the food quality for the larvae likely varies among plant species and cultivars. The cultivation of grass forage species with grains has increasingly been adopted in Brazil, within both no-tillage and crop–livestock integration systems. In this study, we evaluated the performance of *S. frugiperda* larvae on 14 forage cultivars of *Brachiaria*, *Panicum*, and *Cynodon*, which are widely used in integrated cropping systems in Brazil. The biological performance of *S. frugiperda* varied among the cultivars. The larval survival rates were lower on *Panicum maximum* ‘Massai’ and *P. maximum* ‘Tamani’ cultivars. The insects had the highest performance indexes on *Brachiaria brizantha* ‘Paiaguás’, *B. brizantha* ‘Marandu’, and *B. brizantha* ‘Xaraés’ cultivars, followed by *Brachiaria ruziziensis*, previously proposed as a standard grass forage for comparisons with other species. On *P. maximum*, the insect had the lowest performance indexes, with values equal to zero when feeding on the *P. maximum* ‘Massai’ and ‘Tamani’ cultivars. These results will help make management decisions when cultivating grass forage plants in crop production systems in which *S. frugiperda* infestation is of concern.

## 1. Introduction

*Spodoptera frugiperda* (J.E. Smith 1797) (Lepidoptera: Noctuidae) is one of the most important pests in grain production crops such as maize, soybean, cotton, sorghum, rice, and others [1,2,3]. Montezano et al. [4] recorded more than 350 plant species as hosts of *S. frugiperda*, which highlights its great polyphagic ability and capacity to survive in the absence of agricultural crops, since weeds can be a nutritional source for the insect [5]. Furthermore, different host plant and environmental conditions can alter the gut microbiome of the insect and may lead to new approaches to pest management [6].

The diversification of grain production systems with the incorporation of forage plants, mainly Brachiaria grass species (*Brachiaria* syn. *Urochloa*) in rotation or intercropping, has been successfully implemented in different regions of Brazil [7] to be used for pasture or straw formation in the no-tillage system. However, this agricultural practice can extend the availability of host plants of *S. frugiperda*, making the control of this pest even more difficult. There are several advantages of using integrated systems in which forage grasses benefit the soil, mainly by the formation of leftover straw, which contributes to the accumulation of organic matter and the cycling of nutrients through the high production of shoots and roots that bring nutrients present in the depths to the surface [7]. However, this strategy may favor the increase in the population densities of *S. frugiperda* attacking agricultural crops.

*Spodoptera frugiperda* is the most abundant species of Lepidoptera in Brazil, accounting for more than 95% of the caterpillar specimens collected in perennial pastures [8]. Dias et al. [9] showed that the insect may thrive better on young plants of *Brachiaria decumbens* (syn. *Urochloa decumbens)* and *Brachiaria ruziziensis* (syn. *Urochloa ruziziensis*) than on maize, usually considered its preferred host.

Many factors make *S. frugiperda* management difficult, mainly the development of resistance to some *Bt* maize cultivars and chemical insecticides, as well as the intrinsic complexity of grain production systems, in which the entire pest complex needs to be managed (1 and 8). Lv et al. [10] reported that the host plant can change the microbiome of this pest, and this has consequences in its successful management. Hence, it is important to determine the suitability of host plants used within the crop–livestock integration and no-tillage systems to better manage the pest population levels.

Given the possibility of *S. frugiperda* using alternative host plants in the cropping systems, we determined the insect performance on different grass forage plants used as cover crops or intercrops. The information obtained in this study can help predict the risk of *S. frugiperda* infestations in crop production systems integrating the cultivation of grass forages, thus facilitating the process of decision making for better management practices.

## 2. Materials and Methods

### 2.1. Insects and Forage Grasses

The insects were obtained from a rearing laboratory, in which the larvae were kept on an artificial diet [11], and adults were fed with an aqueous honey solution (25 g of sugar; 1 g of ascorbic acid; 1 g maize of glucose in 1.000 mL of distilled water). The insects were kept under controlled conditions in the laboratory, i.e., 26 ± 2 °C, 70 ± 10% relative humidity (RH), 14:10 h light (L)/dark (D). The laboratory rearing was initiated using larvae collected from maize fields (non-transgenic hybrid BRS1030) at Embrapa Milho e Sorgo, Sete Lagoas, Minas Gerais, Brazil. Among the 353 *S. frugiperda* host plant species reported in the literature [4], the 14 most found in agroecosystems in which maize is grown in Brazil were selected (Table 1).

Grasses used to feed the insects were cultivated in 10 m^2^ beds, according to the recommendations for forage cultivation [12]. There was no application of insecticides or seed treatment. They were irrigated according to their needs. For the bioassays, grasses that were 1.0 to 1.5 m tall, approximately 90 to 110 days after germination, were used.

### 2.2. Bioassays

Tests were conducted in the laboratory under controlled conditions (26 ± 2 °C, 70 ± 10% relative humidity (RH), 14:10 h light (L)/dark (D). Plants that reached between 90 to 110 days after germination, in the field, were cut, taken to the laboratory, and cleaned. The following parameters were evaluated: larval survival every 72 h, larval stage survival (total), larval development time, larvae biomass (10 days after hatching), pupae biomass, and fitness index. The larval biomass was measured at 10 days, as this age is adequate to assess differences between treatments, since for the host plants that are more suitable for larval development, the biomass is higher. To evaluate the larvae survival every 72 h, one neonate larva was maintained per cup (50 mL), which was closed with a clear acrylic lid. The cups were disposed on trays, which were kept in a controlled conditioned room. Three sections of tender leaves (approximately 10 cm^2^) from each host plant were used. The leaves were replaced every 72 h, until the end of the larval stage or the death of the insect. The experimental design was completely randomized with five replicates per treatment (14 cultivars). Each replicate consisted of 10 neonate larvae for a total of 50 neonate larvae tested per treatment.

The larval development time was determined from the day the larvae hatched until the first day of pupation. Observations were performed every 72 h. The larvae biomass was determined 10 days after hatching, while the pupae biomass was recorded on the first day of this observation. For both biomass measurements, a precision scale (0.1 mg) was used.

The suitability index (SI) or adaptation index (AI), as used by Boregas et al. (2013) [2] of the 14 host cultivars was calculated. Thus, it was possible to classify and evaluate the performance of *S. frugiperda* on each host. This index is similar to the susceptibility index (SI), which is calculated using the formula: SI = (LS × PB)/(LDT), where: SI = suitability index, LS = larval survival, PB = pupal biomass, and LDT = larval development time. We also calculated the relative suitability index (RSI) using the formula RSI = 100 × (AIh)/(Aim), where Sih = suitability index of *S. frugiperda* in the host in question, and Sim = suitability index of *S. frugiperda* using *Brachiaria ruziziensis* as a control. *Brachiaria ruziziensis* was used because it is one of the most cultivated forages in intensified production systems and the most suitable (suitability index) [9].

### 2.3. Statistical Analyses

For the quantification of differences among and within species, an analysis of variance (ANOVA) was performed through the software R (R Core Team, 2021), using the packages car and agricolae, as well as a subsequent comparison of the treatment means by the Scott–Knott test at a 5% probability level.

## 3. Results

The larval survival of *S. frugiperda* was significantly different (P = 0.0001 and F = 46.155) depending on the host. Higher percentages of larval survival were observed on *B. brizantha* ‘Paiaguás’ (80.0%), ‘Marandu’ (70.5%), ‘Piatã’ (69.5%), ‘Xaraés’ (72.5%) and ‘Ipyporã’ (77.0%), in addition to *B. ruziziensis* (59.0%) (Figure 1a; Table 2). However, on the *Panicum maximum* (syn. *Megathyrsus maximus*) cultivars ‘Tamani’ and ‘Massai’, lower larval survival means were obtained, of 2.5 and 4%, respectively, which did not differ statistically from each other (Figure 1a; Table 2).

For the young stage survival, in other words, for larvae that reached the pupal stage, significant survival differences were also observed (P = 0.0001; F = 56.631). In this case, the highest survival rates continued to be observed on *B. brizantha* ‘Paiaguás’ and ‘Xaraés’, corresponding to 71.5% and 64.5%, respectively. The poor performance of the juvenile phase on *P. maximum* ‘Massai’ and ‘Tamani’ was also highlighted, as on these cultivars, the larvae were not able to complete the pupal stage. Furthermore, a low survival was observed when the larvae were kept on *P. maximum* ‘Tanzania’ and ‘Zuri’, corresponding to 16.0 % and 12%, respectively (Figure 1b).

There was a significant difference for the larval biomass of *S. frugiperda* on the different forage species, which was measured 10 days after their emergence (P = 0.0001; F = 101.680). *B. ruziziensis* allowed a higher larval biomass compared to the other treatments, with a mean of 204.95 mg, 2.4-fold greater than that measured for larvae kept on *B. brizantha* ‘Marandu’ (84.52 mg). The mean values were lower and did not differ from each other on *P. maximum* ‘Massai’ (10.36 mg), ‘Tamani’ (24.91 mg), ‘Tanzânia’ (17.3 mg), and ‘Zuri’ (13.24 mg), and on *B. brizantha* ‘Xaraés’ (26.63 mg) (Figure 2a).

We also found significant differences (P = 0.0001; F = 23.699) for pupal biomass values on the different species and cultivars evaluated. Pupae on *B. ruziziensis* (190.45 mg), *B. brizantha* ‘Marandu’ (180.64 mg), ‘Xaraés’ (171.31 mg), and *Cynodon dactilum* (Bermudagrass) (177.45 mg) presented the highest biomass (Figure 2b). In contrast, when the larvae were kept on *P. maximum* ‘Massai’, only a single pupa of *S. frugiperda* was formed (73.80 mg). On *P. maximum* ‘Mombaça’, ‘Quênia’, ‘Tanzânia’ and ‘Zuri’, the pupae were smaller, with biomass values of 112.30, 107.30, 119.96, and 100.39 mg, respectively.

The larval development time of *S. frugiperda*, evaluated in days, was significantly different (P = 0.0001; F = 38.941), depending on the hosts. *Panicum maximum* ‘Massai’ and ‘Tanzânia’ were the plants that provided the longest larval development times, i.e., 32 and 30.40 days on average, respectively. For *B. brizantha* ‘Marandu’, ‘Paiaguás’ and *C. dactilum* ‘Vaqueiro’, these values were lower, i.e., 18.38, 18.87, and 17.55 days, respectively (Figure 3a). However, on *B. ruziziensis*, the fall armyworm had the shortest larval development time, of 15.42 days (Figure 3a).

Regarding the pre-imaginal development time (young + pupae stages), it was possible to verify that on *P. maximum* ‘Tanzânia’, the larvae extended their development time to an average of 41 days, while on *B. ruziziensis*, the development time as only of 25.4 days (Figure 3b).

The combination of biological variables can be seen in the AI (Figure 4). For *P. maximum* ‘Massai’ and ‘Tamani’, the AI values were equal to zero, as in this case, *S. frugiperda* did not reach the pupae stage. The highest AI values were observed for *B. ruziziensis* and *B. brizantha* ‘Paiaguás’, ‘Marandu’, and ‘Xaraés’.

## 4. Discussion

With more than 350 plants considered as *S. frugiperda* hosts [4], understanding the biology of this polyphagous pest on forage species widely used in Brazil will help our comprehension of how this pest species survives in the field and of its use of plants as alternative hosts. *B. brizantha* ‘Marandu’, for example, was listed [13,14] as the cultivar of a single species that is the most planted in Brazil and in the world. It is noteworthy that other cultivars of *Brachiaria* and *Panicum* occupy about 18.5% of the national territory as cultivated pasture [15]. In this scenario, *S. frugiperda* biology is closely linked to the distribution of these plants under field conditions in many agricultural regions of the country, since the use of these forages as cover crops and in intercropping with grain-producing crops is increasing. In this sense, we evaluated the biological performance of *S. frugiperda* on 14 forage species and obtained a great variation in insect survival according to the forage plant. Our study shows that on cultivars of the genus *Brachiaria*, the insect survival was superior to that on plants of the genus *Panicum*. Dias et al. [9] also showed that the survival of this pest on *Brachiaria*, under field conditions, can be equal to or greater than that found when the larvae were kept on maize, which is considered the insect’s main host. In this case, in addition to the suitability of the host plant to the development of the pest, the authors observed that the large number leaf whorls in the same unit of area provides more shelter for the larvae, thus increasing the larval survival per area unit. Among the cultivars studied, only *B. brizantha* ‘Basilisk’ and *P. maximum* ‘Mombaça’ showed a lower pest survival than the others, showing that even within these genera it is possible to select cultivars with lower suitability for *S. frugiperda*.

The larval biomass at 10 days after emergence provides an indirect measure of the larval growth rate. The greater the larval biomass considering the same time, the more suitable the host plant is. Given this measurement, the host *B. ruziziensis* should be emphasized, since it appeared to allow a higher gain in larval biomass compared to the other host plants. The larval biomass was 2.4-fold larger when the larvae grew on *B. ruziziensis* than the biomass of larvae growing on *B. brizantha* ‘Marandu’, which was the second largest (Figure 5). The data corroborate the study of Dias et al. [9] in which, under greenhouse conditions, *B. ruziziensis* presented a high rate of suitability to *S. frugiperda*. In this sense, when prioritizing the use of this forage in the field, attention must be paid to this feature, and adequate pest monitoring should be carried out.

Pupal biomass is also an important parameter for understanding the biology of lepidopteran pests. According to Pencoe and Martin [16], there is a direct correlation between pupae biomass and adult fertility. In this context, it is expected that the host plants that provide greater pupae biomass will also yield more fertile adults. Hence, cultivars of the *Brachiaria* genus stand out, as they favor *S. frugiperda* development.

Regarding the development time, *S. frugiperda* had longer development times on forages of the genus *Panicum*. The longest time was observed on the cultivar Tanzânia and was almost double that observed on *B. ruziziensis*. In addition, the larvae that were kept on the cultivar ‘Massai’ did not even complete the development phase. Conversely, we observed shorter larval development times on the hosts *B. brizantha* ‘Marandu’, ‘Paiaguás’, *Cynodon dactylum* ‘Vaqueiro’ and, mainly, *B. ruziziensis* (25.4 days on average). This is consistent with the widely accepted premise that the faster the development of the insect, the greater its probability of survival, since it would have less contact with biotic and abiotic mortality factors.

Our results showed that *S. frugiperda* larvae had a better performance on *B. ruziziensis*, followed by *B. brizantha* ‘Xaraés’. These are forage species widely cultivated in diversified systems. Thus, all of these species can serve as alternative host plants in the off-season, especially in maize fields. This is particularly important [17], as the use of maize intercropped with *B. ruziziensis* is predominant in the Santa Fé cultivation system. According to these authors, as Brazil has at least two growing seasons in a year, the main purpose of this system is to increase the amount of straw after harvesting the maize, in such a way that the organic matter also increases, to create a better environment for soybean in the next crop season in summer. *Spodoptera frugiperda* may easily disperse among plants during all phases of larval development, either by wind on silk (ballooning) during the first instars or by walking during all juvenile stages [18]. Therefore, the pest population density can increase in these cropping systems if no management strategy is adopted. These data agree with Falusino et al. [19], who found more damage produced by caterpillars in maize fields intercropping with Brachiaria, than in maize in monoculture.

The grass forages *Brachiaria brizantha* ‘Paiaguás’, ‘Marandu’, ‘Xaraés’ and *B. ruziziensis* were the plants on which *S. frugiperda* had the highest performance index. Boregas et al. [2] found a relative fitness index of 65% with respect to maize for ‘Marandu’ and ‘Tanzânia’ grass. Ribeiro et al. [8] observed a value of 81% for *Cynodon dactylon* ‘Tifton’, and Dias et al. [9] observed that *B. ruziziensis* suitability to *S. frugiperda* was 101% superior to that of maize. This finding led us to select this plant as a reference in the present study. However, we observed an even superior performance on *B. ruziziensis* and *B. brizantha* for both ‘Marandu’ and ‘Paiaguás’ cultivars.

Some studies reported host plant resistance for forage cultivars to *S. frugiperda* [20]. However, *Brachiaria* species have been one of the most used plants in production systems in Brazil [21], whether in no-tillage or in intensified systems. In this context, research comparing the *S. frugiperda* performance on forage species widely cultivated in Brazil may help select better plants to compose the production systems so to minimize the pest infestation. To achieve this goal, the use of genotypes or cultivars that express high resistance levels to *S. frugiperda* is essential, as the cultivars would aid in slowing the larval development and consequently would allow a reduction in the size of subsequent insect populations on successive crops.

In summary, this research showed that the grass forages *P. maximum* ‘Massai’ and ‘Tamani’ are the least suitable hosts for *S. frugiperda* growth and development. On the other hand, *B. brizantha* ‘Marandu’, ‘Paiaguás’ and ‘Xaraés’ are the most suitable host grass forages for the caterpillar. These results will help make management decisions when cultivating grass forage plants in crop production systems in which *S. frugiperda* infestation is of concern.

## Figures and Tables

**Figure 1 insects-13-01139-f001:**
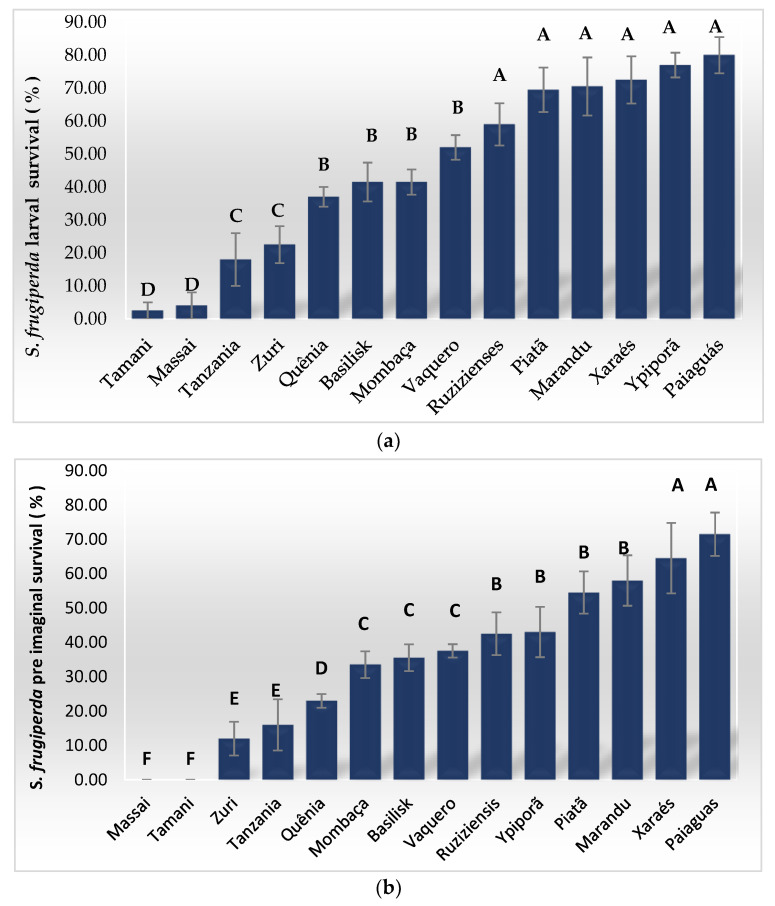
Larval survival (**a**) and pre-imaginal survival (**b**) of *Spodoptera frugiperda* on different forage and cover crops. Means followed by the same letter do not differ from each other by the Scott–Knott test (*p* > 0.05).

**Figure 2 insects-13-01139-f002:**
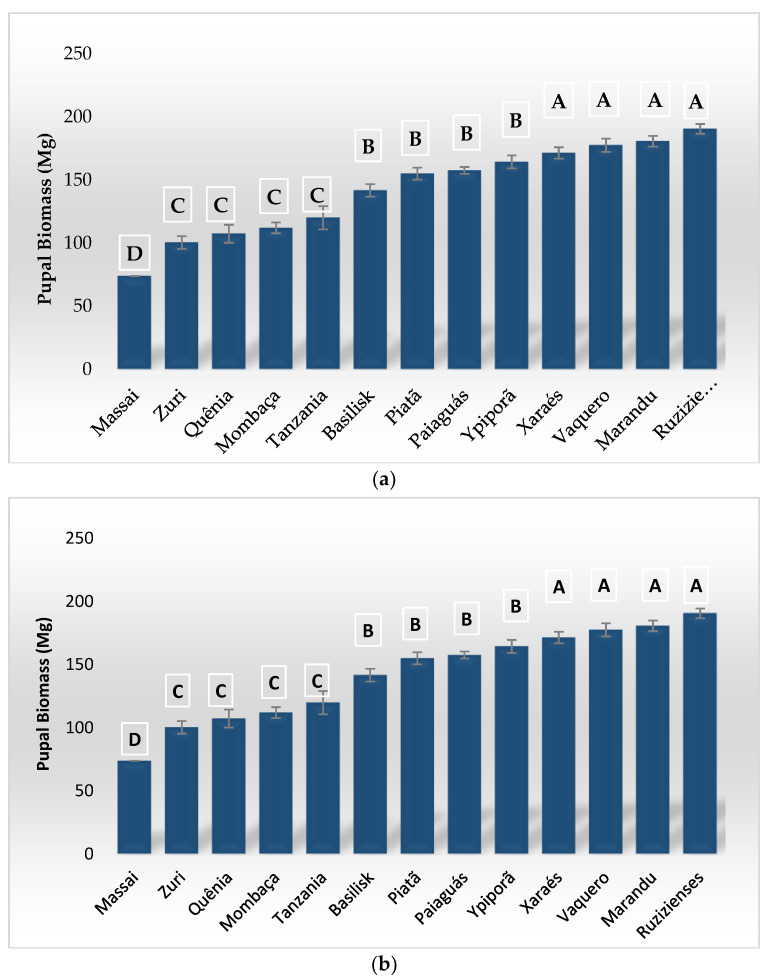
*Spodoptera frugiperda* larvae (10 days old) (**a**) and pupae (**b**) biomass on different forage and cover crops. Means followed by the same letter do not differ from each other by the Scott–Knott test (*p* > 0.05).

**Figure 3 insects-13-01139-f003:**
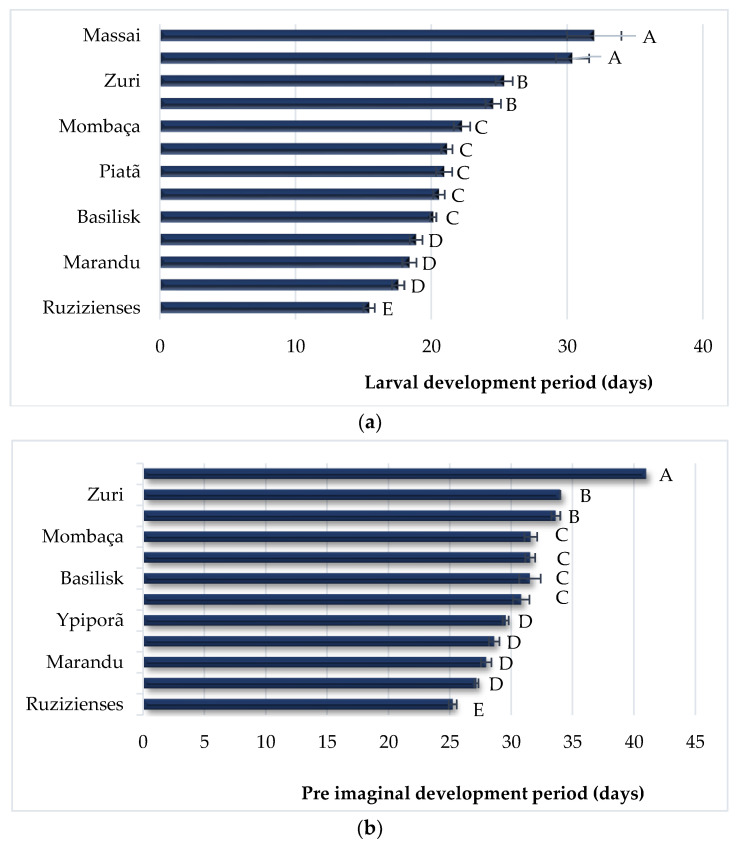
Larval development time (days) (**a**) and pre-imaginal development time (**b**) of *Spodoptera frugiperda* on different forage and cover crops. Means followed by the same letter do not differ from each other by the Scott–Knott test (*p* > 0.05).

**Figure 4 insects-13-01139-f004:**
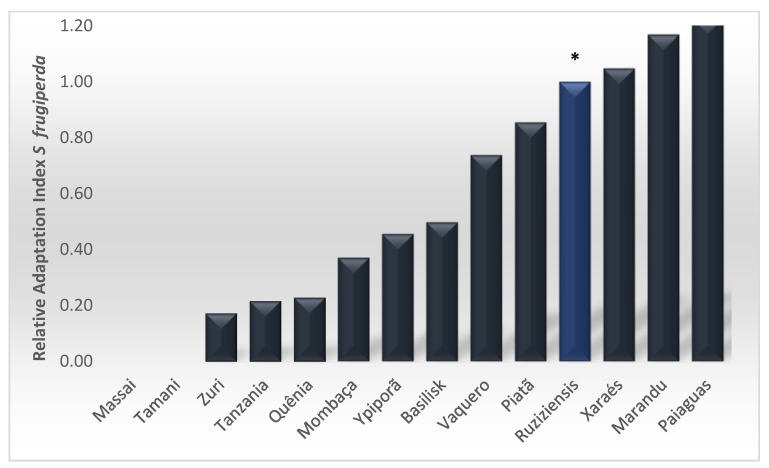
Relative suitability index (RSI) which assesses the larval performance on different alternative hosts, where: SI = larval survival (%) × pupae biomass (mg)/larval development time (days), related to the suitability index (SI) of *Brachiaria ruziziensis **.

**Figure 5 insects-13-01139-f005:**
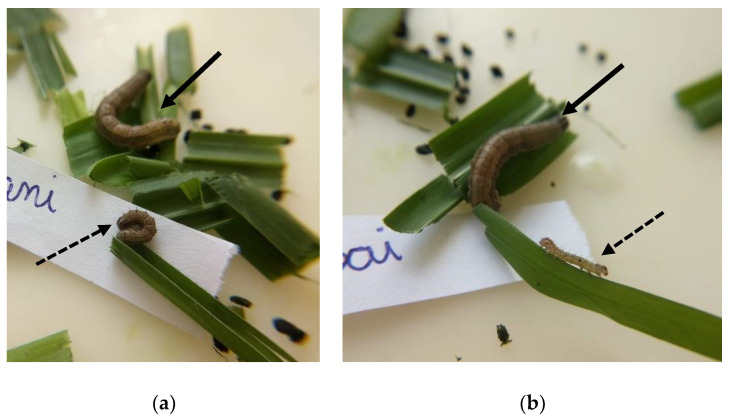
*Spodoptera frugiperda* larvae at 10 days of development on (**a**) *Brachiaria ruziziensis* (bold arrow), *Panicum maximum* ‘Tamani’ (dotted arrow)*,* and (**b**) *Brachiaria ruziziensis* (bold arrow), *Panicum maximum* ‘Massai’ (dotted arrow).

**Table 1 insects-13-01139-t001:** List of grass forage species of the Poaceae family here studied.

Scientific Name	Cultivar	Common Name
*Brachiaria brizantha* (Hochst) Stapf	Marandu	Signal grass
*Panicum maximum* Jacq.	Tanzânia	Guinea grass
*Brachiaria decumbens* Stapf	Basilisk	Signal grass
*Brachiaria brizantha*	BRS Paiaguás	Signal grass
*Brachiaria brizantha*	BRS Xaraés	Signal grass
*Brachiaria* spp.	BRS Ipyporã	hybrid
*Brachiaria brizantha*	BRS Piatã	Signal grass
*Brachiaria ruziziensis* Germain et Evrard	Ruziziensis	Congo signal grass
*Panicum maximum*	Mombaça	Guinea grass
*Panicum maximum*	Massai	Guinea grass
*Panicum maximum*	BRS Zuri	Guinea grass
*Panicum maximum*	BRS Tamani	Guinea grass
*Panicum maximum*	BRS Quênia	Guinea grass
*Cynodon dactylon*	Vaquero	Bermudagrass
Source: Embrapa		

**Table 2 insects-13-01139-t002:** Means (±SE) of biological variables of *Spodoptera frugiperda* larvae fed with different hosts.

Host Plant	Initial Survival (%)	Larval Survival (%)	Biomassa Larval (mg)	Pupa Biomass (mg)	Larval Period (days)	Youth Development Period (days)
Basilisk	4.5 ± 5.9 c	35.5 ± 3.9 c	66.2 ± 4.6 c	141.6 ± 4.9 b	20.1 ± 0.2 c	31.5 ± 0.9 c
Brizantha	70.5 ± 8.8 a	58.0 ± 7.4 b	84.5 ± 5.1 b	180.6 ± 4.3 a	18.4 ± 0.5 d	28.0 ± 0.4 d
Massai	4.0 ± 4.0 e	0.0 ± 0.0 f	10.4 ± 0.9 e	73.8 ± 0.0 d	32.0 ± 2.0 a	0.0 ± 0.0 f
Mombaça	41.5 ± 3.8 c	33.5 ± 3.8 c	39.2 ± 5.7 d	112.0 ± 4.3 c	22.3 ± 0.6 c	31.6 ± 0.5 c
BRS Paiaguás	80.0 ± 5.5 a	71.5 ± 6.3 a	55.5 ± 5.1 c	157.5 ± 2.8 b	18.9 ± 0.5 d	28.6 ± 0.4 d
BRS Piatã	69.5 ± 6.7 a	54.5 ± 6.1 b	48.9 ± 6.3 c	154.9 ± 4.7 b	20.9 ± 0.6 c	30.8 ± 0.7 c
BRS Quênia	37.0 ± 3.0 c	23.0 ± 2.0 d	41.1 ± 2.7 d	107.3 ± 7.1 c	24.6 ± 0.6 b	33.6 ± 0.4 b
Ruziziensis	59.0 ± 6.4 a	42.5 ± 6.2 b	204.9 ± 11.0 a	190.5 ± 3.9 a	15.4 ± 0.4 e	25.2 ± 0.3 e
BRS Tamani	2.5 ± 2.5 e	0.0 ± 0.0 f	24.9 ± 3.1 e	0.0 ± 0.0 f	0.0 ± 0.0 f	0.0 ± 0.0 f
Tanzânia	18.0 ± 8.0 d	16.0 ± 7.5 e	17.3 ± 2.3 e	120.0 ± 9.3 c	30.4 ± 1.2 a	41.0 ± 1.5 a
Vaquero	52.0 ± 3.7 b	37.5 ± 1.9 c	79.2 ± 4.5 b	177.5 ± 5.2 a	17.6 ± 0.5 d	27.2 ± 0.2 d
BRS Xaraés	72.5 ± 7.2 a	64.5 ± 10.3 a	26.6 ± 2.7 e	171.3 ± 4.5 a	21.1 ± 0.4 c	31.6 ± 0.4 c
BRS Ipyporã	77.0 ± 3.7 a	43.0 ± 7.4 b	52.2 ± 3.0 c	164.3 ± 5.2 b	20.6 ± 0.4 c	29.6 ± 0.3 d
BRS Zuri	22.5 ± 5.6 d	12.0 ± 4.9 e	13.2 ± 1.6 e	100.4 ± 5.0 c	25.4 ± 0.6 b	34.0 ± 0.0 b

Averages followed by the same letter, within each column, do not differ, according to the Scott–Knott test, at a 5% probability.

## Data Availability

All data analyzed in this study are included in this article.

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
