# Peer review of "Performance of Fall Armyworm Preimaginal Development on Cultivars of Tropical Grass Forages"

_insects, 2022, doi:10.3390/insects13121139_

Round 1

Reviewer 1 Report

Overall, the manuscript is well written and provides meaningful original information about this significant pest. I have a few general comments and several minor suggestions.

First, the authors must provide a table showing which species each of the cultivars belong to. It can be very confusing trying to keep up with the species and cultivars as written.

Second, the RAI as presented in table 4 is not correct. These numbers should be close to 100 or over 100 for the most suited cultivars. The equation they present is in correct.

Third, it is unusual to find no overlap in significance among means with all of the tests conducted.

Specific comments:

P2, L49: Change relevant to important.

P2, L53: change to read ... in the absence of agricultural crops, ...

P2, L67-70: References to citations need to be in the same font size.

P2, L93: No table 1.

P3, L102-103: Change 'cuted' to 'cut' and 'where cleaned was made' to 'and cleaned'.

P3, L104 and L115: Were observations made every 72 hours or 2 days, unclear.

P3, L105: Be consistent, this is the only use of fitness index, change to adaptation index. Actually I prefer 'suitability index' in that we are discussing the suitability of the plants and their adaptations. It is the insects that can adapt.

P3, L106: Move the sentence for L117 to this spot and insert biomass before measurements.

P3, L107: Delete this sentence.

P3, L114-116: Delete these sentences as they were discussed above.

P3, L123-125: Not sure what is meant by using B. ruziziensis as a control. I assume the AI for maize was the control.

P3, L135: Keep decimals to 2 or 3 places as this implies a higher level of precision.

P3, L138 and 141: Change to Table 2 (But there is no Table 2).

P4, L146: Insert (p>0.05) at end of sentence.

P4, L148: Insert 'survival' before 'differences'.

P4, L160: The figures in figure 2 are reversed. 2a is pupal biomass and 2b is larval. So the references in the text direct the reader to the wrong figures. The same occurs with figure 3.

P5, L161: Where is Tamani? Change 'pupae' to 'pupal'.

P5, L162: Letters above means make no sense and do not correspond to significant differences.

P5, L164: Scott-Knott test not discussed in Materials and Methods.

P6, L187: Insert 'effect of' before 'combination'.

P7, L245: Insert 'as' before 'the use'.

Author Response

We thanks to the reviewers, who made the manuscript much better to read.
The manuscript has been revised considering the comments and suggestions. Tables and photos were added to improve understanding. The graphics have also been redone. The index was evaluated was relativized for one of the plant species (B. ruziziensis). So stay around 1, as described in the text. This understanding was also made by the authors:
1 Boregas, K. G. B., Mendes, S. M., Waquil, J. M., & Fernandes, G. W. (2013). Estádio de adaptação de Spodoptera frugiperda (JE Smith)(Lepidoptera: Noctuidae) em hospedeiros alternativos. Bragantia, 72, 61-70.
2 Dias, A. S., Marucci, R. C., Mendes, S. M., Moreira, S. G., Araujo, O. G., dos Santos, C. A., & Barbosa, T. A. (2016). Bioecology of Spodoptera frugiperda (Smith, 1757) in different cover crops.
3 Barbosa, T. A. N., Mendes, S. M., Rodrigues, G. T., de Aquino Ribeiro, P. E., dos Santos, C. A., Valicente, F. H., & De Oliveira, C. M. (2016). Comparison of biology between Helicoverpa zea and Helicoverpa armigera (Lepidoptera: Noctuidae) reared on artificial diets. Florida Entomologist, 99(1), 72-76.
4 MS Waquil, EJG Pereira, SSS Carvalho, RM Pitta, JM Waquil, SM Mendes Fitness index and lethal time of fall armyworm on Bt corn Pesquisa Agropecuária Brasileira 51, 563-570, 2016

Specific answers

P2, L49: Change relevant to important.

Ok Done

P2, L53: change to read ... in the absence of agricultural crops, ...

Ok Done

P2, L67-70: References to citations need to be in the same font size.

Ok Done

P2, L93: No table 1.

Ok done

P3, L102-103: Change 'cuted' to 'cut' and 'where cleaned was made' to 'and cleaned'.

Ok Done

P3, L104 and L115: Were observations made every 72 hours or 2 days, unclear.

Ok. Corrected

P3, L105: Be consistent, this is the only use of fitness index, change to adaptation index. Actually I prefer 'suitability index' in that we are discussing the suitability of the plants and their adaptations. It is the insects that can adapt.

Ok done

P3, L106: Move the sentence for L117 to this spot and insert biomass before measurements.

Ok Done

P3, L107: Delete this sentence. ok

P3, L114-116: Delete these sentences as they were discussed above. ok

P3, L123-125: Not sure what is meant by using B. ruziziensis as a control. I assume the AI for maize was the control.

It was used because it is one of the most cultivated forage in intensified production systems and the most suitable (suitability index) (9).

Beside the it's the most used in the no-tillage growing, and the most suitable according Dias et al. 2016.

P3, L135: Keep decimals to 2 or 3 places as this implies a higher level of precision.

Ok Done

P3, L138 and 141: Change to Table 2 (But there is no Table 2).

Ok Done

P4, L146: Insert (p>0.05) at end of sentence.

Ok Done

P4, L148: Insert 'survival' before 'differences'. Ok Done

P4, L160: The figures in figure 2 are reversed. 2a is pupal biomass and 2b is larval. So the references in the text direct the reader to the wrong figures. The same occurs with figure 3.

Ok Done

P5, L161: Where is Tamani? Change 'pupae' to 'pupal'.

P5, L162: Letters above means make no sense and do not correspond to significant differences.

P5, L164: Scott-Knott test not discussed in Materials and Methods.

Ok It’s changed

P6, L187: Insert 'effect of' before 'combination'. Ok Done

P7, L245: Insert 'as' before 'the use'. Ok Done

Reviewer 2 Report

The manuscript by Santos and co-authors represents the study on comparative performance of Spodoptera frugiperda on fourteen cultivars of three species of grass forages. The data of survival rate, developmental duration and biomass of larvae and pre-imagines were statistically analyzed. The study has implications for understanding the adaptation of S. frugiperda to local weeds.

Main points:

1.     The title of this manuscript describes the larval stage of the study, but the text contains data for the pupal stage.

2.     When studying larval biomass, why do you choose day 10 larval biomass? We know that the larval stage of Lepidoptera consists of multiple instars. This study does not have specific statistical data on the development time of different instars. At the tenth day of larval development, the instars of S. frugiperda that survive on different weed species may be different. Caterpillars at older instars take significantly more food than caterpillars at younger instars, so there might be a large difference in biomass (Figure 2-b, Ruzizienses). But in the pupal stage, the biomass of S. frugiperda feeding on this cultivar was not so different (Figure 2-a, Ruzizienses). Therefore, the comparison of larval biomass on the tenth day is not suitable, but the comparative analysis of biomass data at the same instar is more appropriate.

3.     The method in the manuscript does not clearly describe the basis for selecting the 14 cultivars, and does not specify which species they belong to. This makes the reader struggle when reading.

4.     The weight of pupae was measured in this study but did not distinguish between males and females. Morphological observations can be used to distinguish between male and female pupae of S. frugiperda, and to calculate the ratio of females and the weight of the pupae of different gender, which may provide more interesting data.

other points

1. The figures need to be modified. It is recommended that all figures be styled the same. In Figures 2 and 3, the cultivars represented on the abscissa were not complete. Figure 2 lacked Tamani and Figure 3 lacked Tamani and Massai.

Figure1: Note in the diagram‘%’position the opposite; Figure2: ‘Ruzizienses’ written ‘Ruzizien...’; Figure 4: ‘Ruzizienses’ written ‘Ruzizien...’

2. Three weed species in the abstract do not have full scientific names.

3. Line 100-103: “…Plants that reached the develop-101 ment stage defined for this study (4-5 leaves) were cuted and taken to the laboratory 102 where cleaned was made. ” The definition of developmental stage leaves is too vague.

4. It is recommended to mark the language at the end of non-English references

Author Response

  1. The title of this manuscript describes the larval stage of the study, but the text contains data for the pupal stage.

        reply: Ok – We changed larvae to preimaginal

  1. When studying larval biomass, why do you choose day 10 larval biomass? We know that the larval stage of Lepidoptera consists of multiple instars. This study does not have specific statistical data on the development time of different instars. At the tenth day of larval development, the instars of  frugiperdathat survive on different weed species may be different. Caterpillars at older instars take significantly more food than caterpillars at younger instars, so there might be a large difference in biomass (Figure 2-b, Ruzizienses). But in the pupal stage, the biomass of S. frugiperda feeding on this cultivar was not so different (Figure 2-a, Ruzizienses). Therefore, the comparison of larval biomass on the tenth day is not suitable, but the comparative analysis of biomass data at the same instar is more appropriate.

         reply:A LARVAL BIOMASS – INSIDE THE TEXT NOW -Larval biomass was measured at 10 days, as this age is adequate to assess the difference between treatments, since for the host plants that are more suitable for larval development, the biomass is higher.   

          This methodology of measuring the biomass at 10 days is exactly to observe the larval growth among the different host plants. The more suitable the host plant, the greater the biomass. This can be seen in the photos that were added (figures). The measurement of biomass within a given instar would not be adequate to show this difference. The purpose is exactly that measure the "cut" time - larval biomass - which shows the suitability of the host plant and the other is the "cut in the stage of development - pupal biomass - regardless of how many days the larva took to develop, if it has "reached" the pupal stage which will be measured at that instar.

           Another issue, we are not dealing here with weed local species, but with forage plants widely used in tropical farming systems, no-tillage systems, or pasture.

  1. The method in the manuscript does not clearly describe the basis for selecting the 14 cultivars, and does not specify which species they belong to. This makes the reader struggle when reading.

       reply:Please, see Table I and the Introduction.

  1. The weight of pupae was measured in this study but did not distinguish between males and females. Morphological observations can be used to distinguish between male and female pupae of  frugiperda, and to calculate the ratio of females and the weight of the pupae of different gender, which may provide more interesting data.

        reply:The experiments are finished now, however, the mean without distinguishing males and females was done for all treatments.

other points

  1. The figures need to be modified. It is recommended that all figures be styled the same. In Figures 2 and 3, the cultivars represented on the abscissa were not complete. Figure 2 lacked Tamani and Figure 3 lacked Tamani and Massai.

        reply: Figure1: Note in the diagram‘%’position the opposite; Figure2: ‘Ruziziensis’ written ‘Ruzizien...’; Figure 4: ‘Ruziziensis’ written ‘Ruzizien...’

  1. Three weed species in the abstract do not have full scientific names.

         reply: Ok it’s done in the text

  1. Line 100-103: “…Plants that reached the develop-101 ment stage defined for this study (4-5 leaves) were cuted and taken to the laboratory 102 where cleaned was made. ” The definition of developmental stage leaves is too vague.

          reply: Ok  Its changed in text

  1. It is recommended to mark the language at the end of non-English references

         reply: Ok it’s done in the text

Round 2

Reviewer 2 Report

  • The title of the revised edition has not been changed, and Tables 1 and 2 are not included in the manuscript.

Author Response

Dear Reviewer, Changes have been made: We changed larval to preimaginal in the title and insert the table1 and 2.
Thanks!

Round 3

Reviewer 2 Report

-

Author Response

Dear editor,
We reviewed the manuscript and the attached file about authors.

Thanks

Name

Affiliation

author (1)

Marcos Vinicius Campos Dos Santos

Universidade Federal de São João Del Rei, Campus de Sete Lagoas, Rodovia MG-424, Km 47, Bairro Indústrias, Caixa Postal 56, Sete Lagoas, MG, 35701-970, Brazil

author (2)

Priscilla Tavares Nascimento

Embrapa Milho e Sorgo, Rodovia MG 424, Km 45, P. O. Box 285, Sete Lagoas, MG, 35701-970, Brazil

author (3)

Patrick Ferreira Lima

Universidade Federal de São João Del Rei, Campus de Sete Lagoas, Rodovia MG-424, Km 47, Bairro Indústrias, Caixa Postal 56, Sete Lagoas, MG, 35701-970, Brazil

author (4)

Maria Lucia Simeone

Embrapa Milho e Sorgo, Rodovia MG 424, Km 45, P. O. Box 285, Sete Lagoas, MG, 35701-970, Brazil

author (5)

Rosangela Maria Simeao

Embrapa Gado de Corte, Grupo de Produção Vegetal, Av. Rádio Maia, 850, Bairro Vila Popular, 79106-550, Campo Grande, MS, Brazil.

author (6)

Alexander Machado Auad

Embrapa Gado de Leite, laboratório de Entomologia, Embrapa Gado de leite. Juiz de Fora, MG Brazil

author (7)

Ivênio Rubens de Oliveira

Embrapa Milho e Sorgo, Rodovia MG 424, Km 45, P. O. Box285, Sete Lagoas, MG, 35701-970, Brazil. Fone +55: (31) 3027-1136

author (8

Simone Martins Mendes 

Embrapa Milho e Sorgo, Rodovia MG 424, Km 45, P. O. Box285, Sete Lagoas, MG, 35701-970, Brazil. Fone +55: (31) 3027-1136
